# Quantification of Type I Interferon Inhibition by Viral Proteins: Ebola Virus as a Case Study

**DOI:** 10.3390/v13122441

**Published:** 2021-12-04

**Authors:** Macauley Locke, Grant Lythe, Martín López-García, César Muñoz-Fontela, Miles Carroll, Carmen Molina-París

**Affiliations:** 1Department of Applied Mathematics, School of Mathematics, University of Leeds, Leeds LS2 9JT, UK; mmmwl@leeds.ac.uk (M.L.); grant@maths.leeds.ac.uk (G.L.); m.lopezgarcia@leeds.ac.uk (M.L.-G.); 2Bernhard Nocht Institute for Tropical Medicine, Bernhard Nocht Straße 74, 20359 Hamburg, Germany; munoz-fontela@bnitm.de; 3German Center for Infection Research (DZIF), Partner Site Hamburg, Bernhard Nocht Straße 74, 20359 Hamburg, Germany; 4Wellcome Centre for Human Genetics, Nuffield Department of Medicine, University of Oxford, Oxford OX3 7BN, UK; miles.carroll@ndm.ox.ac.uk; 5T-6, Theoretical Biology and Biophysics, Theoretical Division, Los Alamos National Laboratory, Los Alamos, NM 87545, USA

**Keywords:** type I interferon, virus, Ebola, mathematical model, immune response inhibition, antagonist protein

## Abstract

Type I interferons (IFNs) are cytokines with both antiviral properties and protective roles in innate immune responses to viral infection. They induce an antiviral cellular state and link innate and adaptive immune responses. Yet, viruses have evolved different strategies to inhibit such host responses. One of them is the existence of viral proteins which subvert type I IFN responses to allow quick and successful viral replication, thus, sustaining the infection within a host. We propose mathematical models to characterise the intra-cellular mechanisms involved in viral protein antagonism of type I IFN responses, and compare three different molecular inhibition strategies. We study the Ebola viral protein, VP35, with this mathematical approach. Approximate Bayesian computation sequential Monte Carlo, together with experimental data and the mathematical models proposed, are used to perform model calibration, as well as model selection of the different hypotheses considered. Finally, we assess if model parameters are identifiable and discuss how such identifiability can be improved with new experimental data.

## 1. Introduction

We are exposed to a diversity of pathogens throughout our lives. Our immune system, both its innate and adaptive arms, has developed molecular and cellular mechanisms to sense, prevent and respond to such infections. Cells in the first line of protection, the innate immune system, are equipped with pattern-recognition receptors (PRRs) that sense pathogen-associated molecular patterns (PAMPs), such as microbial products (e.g., viral RNA) [1]. Activation of PRRs in infected cells leads to secretion of type I interferon (IFN), the main antiviral cytokine [2,3,4,5]. Binding of type I IFN to its receptor, in turn, induces the transcription of a family of interferon-stimulated genes (ISGs), whose protein products have both antiviral activity and immuno-modulatory effects [2,3,6].

The survival of a viral population in a host depends on viruses replicating and avoiding intra-cellular defences. Many viruses have developed strategies to evade immune detection [7], and thus, antagonise these defences [7]. There exist a great diversity of such viral strategies and here we will only consider those mechanisms which interfere with intra-cellular pathways to either regulate type I IFN secretion or type I IFN signalling. For instance, influenza, hepatitis C virus, vaccinia or herpes simplex virus [3,8] all inhibit type I IFN responses.

Interferons are a family of cytokines which not only regulate cellular growth but also have antiviral properties and immuno-modulatory activity [1]. They elicit an antiviral state defined by intra-cellular antimicrobial programmes, which require the induction of IFN stimulated genes. Interferons can be broadly classified into type I and type II. Type I interferons (or viral IFNs), which are secreted by infected cells, include IFN-α, IFN-β, IFN-ω and IFN-τ. Most cell types can produce IFN-β, which is the best characterised member of the family. Hematopoietic cells produce IFN-α. IFN-γ is a type II IFN and is only produced by certain immune (professional) cells, such as natural killer cells, CD4${}^{+}$ (helper) and CD8${}^{+}$ (cytotoxic) T cells [7].

Filoviruses, such as Ebola virus (EBOV) and Marburg virus, encode viral proteins with the ability to counteract type I IFN responses in order to replicate in an efficient manner and minimise the therapeutic antiviral power of IFNs. These type I IFN antagonist proteins, or viral antagonistic proteins (VAPs), are essential to guarantee viral replication, prevent the type I IFN-induced antiviral state in infected and bystander cells, as well as impair the ability of antigen presenting cells to initiate adaptive immune responses [9]. This ability of filoviruses to replicate ‘unchecked’ by the host innate antiviral response can partly account for their lethality of infection. Early innate immune evasion facilitates fast and excessive viral replication, which in turn, activates a damaging host immune response [2]. Unfortunately, filoviruses are not the only viral family to actively avoid immune surveillance. Other examples include influenza A virus, hepatitis B virus, and Bunyaviruses, such as Crimean-Congo haemorrhagic fever (CCHFV) or Rift Valley fever viruses [10].

It is important to note that there exists stark contrast, and even conflicting evidence, between responses to in vivo and in vitro models of infection. In EBOV infection, for example, type I IFN production is abrogated after three days post-infection in in vitro infection, yet for in vivo infection, type I IFN cytokines are secreted during the entire infective period [11,12,13]. In this paper, we develop mathematical models of the intra-cellular molecular processes that are known to antagonyse type I IFN production by viral proteins [2,6]. Existing mathematical models of intra-cellular production of type I IFN have many parameters that are difficult to estimate, or do not account for viral protein antagonism of PRR pathways [14,15,16]. There are also models that describe inter-cellular interactions via IFN α receptors [17]. Our goal is to model mechanisms of upstream and downstream viral protein antagonism, and to provide a case study applied to Ebola virus. We calibrate our models with clinical data from in vivo Ebola infection of rhesus macaques [12]. Model selection allows us to compare different biological hypotheses. Our approach could serve as an example to characterise and quantify inhibition of type I IFN secretion by other pathogens, such as SARS-COV-2 virus [18,19], dengue and West Nile viruses [20], hantaviruses [21], and bunyaviruses [22].

## 2. Materials and Methods

### 2.1. Mathematical Models of Type I IFN Inhibition by Viral Protein

There exist fourteen different IFN-α genes but only one IFN-β gene and one IFN-γ gene. IFNs mediate their effect via interactions with type-specific receptors. IFN receptors do not have enzymatic activity but they initiate a signalling cascade (or pathway) which results in the transcription of a large number (of the order of a hundred) of IFN-stimulated genes (ISGs) [7]. In what follows we shall focus on IFN-β secretion (or synthesis) and not on IFN receptor signalling pathways.

IFN-β synthesis requires several transcription factor (TF) complexes, such as NF-κB or ATF, and IFN-regulated factors (IRFs). These TFs are activated by phosphorylation of serine residues. Activation of IRFs is triggered by viral infection, most likely, by the production of viral RNA and other virus specific signals. Expression of IFN-β seems to be induced early in a viral infection through the activation of IRF3, which is constitutively expressed, and does not require to be transcriptionally activated via IFN receptor-mediated signalling and the JAK/STAT pathway. IRF7, on the other hand, is not constitutively expressed, and in fact, IFN-β provides the initial signal that allows IRF7 to be synthesised, which in turn leads to the expression of the full spectrum of IFNs and ISGs [1,7].

Type I IFN production is induced after the sensing of microbial products, such as PAMPs, by PRRs [1]. PRRs include RIG-I-like receptors (RLRs), such as RIG-I. RLRs are cytoplasmic RNA helicases that detect viral RNAs and promote IFN responses. In the case of filovirus infection RIG-I is the most relevant RLR. The first step in the process is the activation of RIG-I by immuno-stimulatory RNAs, such as viral RNA with 5′ triphosphates and double stranded (ds) RNA, produced during filovirus replication. The activation of RIG-I is mediated by its binding to RNA or dsRNA. Activated RIG-I activates the TBK1 kinase, which in turn phosphorylates the transcription factor IRF3. This event promotes the nuclear accumulation of IRF3 to initiate the expression of type I IFNs. Once expressed, type I IFNs secreted from cells can bind its hetero-dimeric receptor, the type I IFN receptor (or IFNR), which will activate the JAK-STAT signalling pathway. This will lead to the induction of ISG expression, which in turn triggers an antiviral state that renders cells refractory to viral infection [2,9].

In the specific case of EBOV, which rather effectively counters this antiviral defence by blocking the production of type I IFNs, this inhibition seems to depend on the observation that EBOV genomic RNA can activate RIG-I by impairing the ability of the host cell to detect the presence of viral products in the cytosol. EBOV VAP VP35 mediates this subversion of cytosolic sensing and different mechanisms of inhibition have been observed both upstream and downstream of the RLR-induced signalling cascade. Downstream targets of VP35 are protein kinases IKKε and TBK1, since in fact, VP35 inhibits phosphorylation of both of them [2,9]. These family of protein kinases coordinates the activation of interferon-regulatory factor (IRF) proteins [2,6]. Upstream inhibitory activities are related to the dsRNA binding ability of VP35. There is also evidence that VP35 can bind to a protein called PACT, which in turn can bind to activated RIG-I. We note that VP35 can bind to PACT in the absence of dsRNA. PACT is an activator of protein kinase R (PKR), with PKR an IFN-induced dsRNA-activated kinase. Finally, VP35 has shown ability to inhibit IFN-induced antiviral proteins, and thus, can elicit downstream suppression of RIG-I signalling [2,9,23,24].

We conclude with a reflection on the fact that the mechanisms of VP35 induced inhibition of type I IFN pathways have not been completely elucidated yet [2,9,23,24]. Thus, in this section we aim to propose three different mathematical models of VAP inhibition of type I IFN expression, each of them considering different molecular mechanisms of suppression, so that together with data from EBOV in vivo infection, we can compare model predictions and carry out parameter calibration.

#### 2.1.1. A First Model of Type I IFN Inhibition by VAP

A first mathematical model can be introduced to characterise the inhibition of type I IFN secretion by viral antagonistic protein (VAP), and which considers the role of the following proteins: RIG-I, viral RNA, VAP and TBK1, a protein kinase which coordinates the activation of interferon-regulatory factor (IRF) proteins [2,6]. We note that RIG-I is an important cytosolic PRR, and that the specific VAP will depend on the virus under consideration; for instance, if the virus is EBOV, then VAP is VP35, and in the case of Bunyaviruses VAP is the non-structural protein NSs [10]. We denote RIG-I by *R*, viral RNA by *D*, VAP by *V*, and TBK1 by *B*. Driven by current experimental evidence, we propose the following reactions [2,6] shown in Figure 1. The first reaction describes RIG-I and viral RNA binding to form a RIG-I:RNA complex (R:D) with rate $k_{R}$, and unbinding with rate $q_{R}$. We shall assume mass action kinetics in what follows. The second reaction describes VAP and viral RNA binding to form a VAP:RNA complex (*V*:*D*) with rate $k_{V}$, and unbinding with rate $q_{V}$. The last reaction describes activation, i.e., phosphorylation, of TBK1 with rate $k_{B}$, and de-activation (de-phosphorylation) with rate $q_{B}$. We denote the activated *B* molecule by $B^{*}$. Let us refer to this model as ‘model 1’. We denote by $n_{R}$, $n_{D}$, $n_{V}$, and $n_{B}$, the per cell total number (or copy number) of RIG-I proteins, viral RNA molecules, VAP proteins and TBK1 proteins, respectively. We neglect protein degradation or synthesis, so that for the timescales considered, the total number of molecules of a given species is conserved. We denote the number of RIG-I:RNA (or R:D) complexes at time t≥0 by $n_{RD}\left(t\right)$, the number of VAP:RNA (or *V*:*D*) complexes by $n_{VD}\left(t\right)$, and the number of activated TBK1 (or $B^{*}$) complexes by $n_{B^{*}}\left(t\right)$ [25]. Conservation of copy numbers implies that the number of free RIG-I molecules at any given time is given by $n_{R}-n_{RD}\left(t\right)$, the number of free viral RNA is given by $n_{D}-n_{RD}\left(t\right)-n_{VD}\left(t\right)$, the number of free VAP molecules is given by $n_{V}-n_{VD}\left(t\right)$, and the number of unphosphorylated TBK1 molecules is given by $n_{B}-n_{B^{*}}\left(t\right)$. Based on the reactions described in Figure 1, we now can write the following system of ODEs to describe the dynamics of the mathematical model: (1)$$\begin{array}{cc} \frac{dn_{RD}}{dt} & =k_{R}\left(n_{D}-n_{RD}-n_{VD}\right)\left(n_{R}-n_{RD}\right)-q_{R}n_{RD},\\ \frac{dn_{VD}}{dt} & =k_{V}\left(n_{D}-n_{RD}-n_{VD}\right)\left(n_{V}-n_{VD}\right)-q_{V}n_{VD},\\ \frac{dn_{B^{*}}}{dt} & =\frac{k_{B}\left(n_{B}-n_{B^{*}}\right)n_{RD}}{\kappa_{V}+\left(n_{V}-n_{VD}\right)}-q_{B}n_{B^{*}}. \end{array}$$

These equations encode both upstream and downstream viral antagonism in the IFN type I secretion pathway [2,6,26,27]. Firstly we have RNA silencing carried out by VAP (or *V*). This results in a competition process for viral RNA with RIG-I, since both VAP and RIG-I can bind to viral RNA. This is the upstream mechanism of viral antagonism to inhibit type I IFN expression. Additionally, the third ODE (see Equation (Equation 1)) for $n_{B^{*}}$ describes activation of *B* in the presence of RIG-I:RNA complexes, $n_{RD}$. This equation includes the antagonistic effect of VAP in the phosphorylation of TBK1, encoded in the denominator, $\kappa_{V}+\left(n_{V}-n_{VD}\right)$, which implies free VAP lowers the ‘effective’ rate of TBK1 phosphorylation, with a carrying capacity $\kappa_{V}$. In this way, model 1 incorporates a downstream inhibitory mechanism as well.

#### 2.1.2. A Second Model of Type I IFN Inhibition by VAP: PACT Protein

The model shown in Figure 1 characterises two key aspects of the role of VAP: upstream antagonism with RIG-I when binding to viral RNA and downstream antagonism in the activation of TBK1. Yet, there are additional strategies that viruses explore to inhibit type I IFN secretion. Protein activator of the interferon-induced protein kinase (PACT) has been identified as a secondary activator of RIG-I during viral infections [23,28]. This molecule provides an additional activation route for RIG-I, and thus, a boost to type I IFN induction (see Figure 2). However, just like RNA silencing (i.e., the binding of VAP to viral RNA to compete with RIG-I), many viruses have mechanisms to inhibit the interaction of PACT with RIG-I. Influenza virus protein NS1, Ebola VP35 and MERS-CoV protein 4a (p4a) have all been identified to cause antagonism of the interaction between PACT and RIG-I [23,29,30]. Therefore, to include this second viral strategy of innate immunity inhibition we propose a second model which includes PACT, and all the other molecular species and reactions of model 1. *R*, *D*, *V* and *B* retain their previous definitions (see Section 2.1.1), and we denote PACT by *P*. Based this experimental evidence we propose the following set of reactions illustrated in Figure 2. The first, second and final reactions remain unchanged from model 1 (see Figure 1). The third reaction includes the binding of RIG-I and PACT to form a RIG-I:PACT complex (*R*:*P*) with rate $k_{P}$, and dissociation rate $q_{P}$. The final new reaction (fourth reaction) includes the binding of VAP to PACT to form a VAP:PACT complex (*V*:*P*) with binding rate $k_{M}$, and unbinding rate $q_{M}$. We shall refer to this as ‘model 2’. Let us denote by $n_{R}$, $n_{D}$, $n_{P}$, $n_{V}$, and $n_{B}$, the per cell total number of RIG-I, viral RNA molecules, PACT, VAP and TBK1, respectively. As previously described we neglect protein degradation and synthesis, so that the total number of molecules for each protein species is conserved. We denote the total number of RIG-I:RNA (*R*:*D*) complexes at time t≥0 by $n_{RD}\left(t\right)$, VAP:RNA (*V*:*D*) by $n_{VD}\left(t\right)$, RIG-I:PACT (*R*:*P*) by $n_{RP}\left(t\right)$, VAP:PACT (*V*:*P*) by $n_{VP}\left(t\right)$ and activated TBK1 ($B^{*}$) by $n_{B^{*}}\left(t\right)$. As before we introduce the following set of differential equations:(2)$$\begin{array}{cc} \frac{dn_{RD}}{dt} & =k_{R}\left(n_{D}-n_{RD}-n_{VD}\right)\left(n_{R}-n_{RD}-n_{RP}\right)-q_{R}n_{RD},\\ \frac{dn_{VD}}{dt} & =k_{V}\left(n_{D}-n_{RD}-n_{VD}\right)\left(n_{V}-n_{VD}-n_{VP}\right)-q_{V}n_{VD},\\ \frac{dn_{RP}}{dt} & =k_{P}\left(n_{R}-n_{RD}-n_{RP}\right)\left(n_{P}-n_{RP}-n_{VP}\right)-q_{P}n_{RP},\\ \frac{dn_{VP}}{dt} & =k_{M}\left(n_{P}-n_{RP}-n_{VP}\right)\left(n_{V}-n_{VD}-n_{VP}\right)-q_{M}n_{VP},\\ \frac{dn_{B^{*}}}{dt} & =\frac{k_{B}\left(n_{RD}+n_{RP}\right)\left(n_{B}-n_{B^{*}}\right)}{\kappa_{V}+n_{V}-n_{VD}-n_{VP}}-q_{B}n_{B^{*}}. \end{array}$$

These newly defined equations include the previous two (upstream and downstream) inhibition mechanisms, as well as the additional role of PACT to activate RIG-I, in the form of the complex $n_{RP}$, and the upstream inhibition mechanism encoded in the competition between *R* and *V* to bind to *P*. These are reflected in the fifth equation of Equation (Equation 2) for activated TBK1 ($n_{B^{*}}$): the phosphorylation of TBK1, with rate $k_{B}$, is proportional to the total amount of activated RIG-I molecules, $n_{RD}+n_{RP}$, and is inhibited by the presence of VAP, $n_{V}-n_{VD}-n_{VP}$, with carrying capacity $\kappa_{V}$.

#### 2.1.3. A Third Model of Type I IFN Inhibition by VAP: PKR Signalling Pathway

We have proposed two models which examine the effects of VAP on RIG-I induced type I interferon induction. However, as discussed in our introduction, other PRRs exist. We now consider one such alternative pathway. Protein kinase R (PKR) also binds to viral RNA, and the resulting bound complex, *A*:*D* in Figure 3, induces the type I IFN secretion pathway [31,32]. This pathway can also be hijacked by viruses. In fact, influenza, herpes simplex 1 and Ebola viruses have been observed to inhibit the PKR pathway [33,34]. In light of the current experimental evidence, we introduce a third and final mathematical model. In this model the complex, *A*:*D* (see Figure 3), plays the role performed by phosphorylated TBK1 in models 1 and 2, as the downstream element in the pathway to induce type I IFN secretion. That is, this model does not consider the RIG-I pathway, but it describes the PKR one. Yet, we consider the role of RIG-I in sequestering viral RNA from both VAP and PKR, and the role of RIG-I in sequestering PACT from VAP.

We consider molecules *R*, *D*, *V* and *P*, as in models 1 and 2. We now introduce *A* to represent PKR. We retain the first four reactions shown in Figure 2 to keep competition for viral RNA between VAP, RIG-I and PKR. We also keep the previous reactions that involve free VAP molecules. The final set of reactions considered, with rates $k_{A}$ and $q_{A}$, respectively, are presented in Figure 3. Our fifth reaction is that of PKR binding to viral RNA, with rate $k_{A}$, resulting in the formation of phosphorylated PKR:RNA (*A*:*D*) complex. This complex can become unphosphorylated and disassociate with rate $q_{A}$. In the presence of VAP, PKR can be actively dephosphorylated and disassociate with rate $q_{AV}$. Thus, in this model, there exists a new viral strategy to inhibit innate recognition via type I IFN. The ODEs for this model are described in Equation (Equation 3). Variables $n_{RD}\left(t\right),n_{VD}\left(t\right),n_{RP}\left(t\right)$ and $n_{VP}\left(t\right)$ represent the same complexes as in model 2. The new variable, which describes complex *A*:*D*, $n_{AD}\left(t\right)$, now represents the number of phosphorylated PKR:RNA complexes. We refer to this model as ‘model 3’.
(3)$$\begin{array}{cc} \frac{dn_{RD}}{dt} & =k_{R}\left(n_{R}-n_{RD}-n_{RP}\right)\left(n_{D}-n_{RD}-n_{VD}-n_{AD}\right)-q_{R}n_{RD},\\ \frac{dn_{VD}}{dt} & =k_{V}\left(n_{V}-n_{VD}-n_{VP}\right)\left(n_{D}-n_{RD}-n_{VD}-n_{AD}\right)-q_{V}n_{VD},\\ \frac{dn_{RP}}{dt} & =k_{P}\left(n_{R}-n_{RD}-n_{RP}\right)\left(n_{P}-n_{RP}-n_{VP}\right)-q_{P}n_{RP},\\ \frac{dn_{VP}}{dt} & =k_{M}\left(n_{V}-n_{VD}-n_{VP}\right)\left(n_{P}-n_{RP}-n_{VP}\right)-q_{M}n_{VP},\\ \frac{dn_{AD}}{dt} & =k_{A}\left(n_{A}-n_{AD}\right)\left(n_{D}-n_{RD}-n_{VD}-n_{AD}\right)-\left[q_{A}+q_{AV}\left(n_{V}-n_{VD}-n_{VP}\right)\right]n_{AD}. \end{array}$$

We remind the reader that all reactions in this model are described by mass action kinetics, except the one proportional to $q_{AV}$. In this case, and in order to model the de-phosphorylation and disassociation enhancement caused by the VAP, we have added a term proportional to the number of free VAP molecules, $n_{V}-n_{VD}-n_{VP}$. Finally, we note that in this model TBK1 is assumed to be either non-functional as part of the signalling pathway, or insufficiently stimulated, to contribute to production of type I IFN.

As discussed in the introduction, the three models proposed in this section can potentially be applied to a number of different viruses, which exhibit similar antagonism strategies. In what follows we will restrict our study to Ebola virus. In the case of EBOV, the antagonistic viral protein to type I IFN secretion pathways is called VP35 [26,35,36]. We will make use of approximate Bayesian computation to perform model calibration, as well as model selection to identify and quantify which hypothesis (viral inhibition strategy) better explains the data set. In the next section we describe the nature of the data set that we have used to carry out model selection and calibration.

### 2.2. Transcriptomic Data

Kotliar et al. made use of single-cell transcriptomics and CyTOF-based single-cell protein quantification to characterise peripheral immune cells during EBOV infection in rhesus monkeys [12]. Their analysis allowed them to conclude that the interferon response is suppressed in infected cells. We make use of their transcriptomic data set for parameter calibration of the mathematical models introduced in the previous sections (see Section 2.1). In particular, we perform approximate Bayesian computation sequential Monte Carlo (ABC-SMC) [37,38]. Eighteen non-human primates (NHPs) were exposed to the EBOV/Kikwit isolate (*Kikwit-9510621*) diluted to a target concentration of $10^{3}$ plaque forming units (PFU) in a volume of 1 mL/doses [12]. Two baseline blood samples were collected between 0 and 14 and 14–30 days prior to infection. Post-infection (PI) clinical observations and whole blood collection were carried out daily until day eight PI (see Figure 1 in Ref. [12]). A total of $19\times 20^{3}$ genes were tested with single cell RNA-sequencing. We shall make use of transcript counts ($10^{4}$) for IFN-β to perform parameter calibration for each mathematical model [12]. The data are summarised in Table 1.

### 2.3. Bayesian Inference

Reports of parameter values for this biological system are limited, thus, we estimate parameters with the approximate Bayesian computation sequential Monte Carlo algorithm (ABC-SMC) [37]. Posterior distributions of model parameters are obtained through sequential application of the ABC algorithm for *K* iterations [38]. For this algorithm we define a prior distribution for the first iteration, where subsequent iterations use posterior distributions from the one preceding as a prior. We assume all parameters follow a uniform prior distribution as defined in Table 2. For each iteration we require a threshold value, $\varepsilon_{k}>0$, k=1,…,K, perturbation kernel, and distance measure, *d* [37]. To maximise exploration of parameter ranges uniform distributions are taken from the exponent base 10. Given a set of parameters, θ, for any of the three mathematical models introduced in the previous sections, we define a Euclidean distance measure to be
$$d\left(x,y|\theta \right)=\sqrt{\sum_{t\in T}{\left(x\left(t\right)-y\left(t\right)\right)}^{2}}\;,$$
where T is the set of time points in the data set, x(t) denotes the output from the mathematical model at time *t* for parameters θ, and y(t) represents experimental data at time *t*. We note that in this case, x(t) will be the variable representing phosphorylated TBK1 for models 1 and 2 ($n_{B^{*}}\left(t\right)$), while for model 3 we consider phosphorylated PKR ($n_{AD}\left(t\right)$). We assume a linear relationship between transcripts and protein numbers, which is of course a good approximation. Our first iteration threshold $\varepsilon_{1}$ is defined as the median of $10^{6}$ initial realisations via ABC with prior samples obtained from the distributions in Table 2. We then define threshold $\varepsilon_{k}$ as the median distance from iteration k−1. Our perturbation kernel will be uniform and will be used to perturb the parameter values during each iteration. Each iteration will be run until a total $n=25\times 10^{2}$ parameter sets are accepted.

## 3. Results

### 3.1. Identifiability of Model Parameters

Before carrying out model calibration it is important to study the structural identifiability of the parameters. Since many of the parameters of our mathematical models have not been previously determined, we must understand whether we can estimate their values given our limited data set [41]. Examining model 1 we find that $q_{B}$, $k_{B}$ and $n_{B}$ are locally identifiable parameters. This is due to the fact that the variable $n_{B^{*}}\left(t\right)$, phosphorylated TBK1, is the model output which we compare to data. Using the Structural Identifiability Analyzer (SIAN) we find that $q_{V}$ is locally identifiable [42]. Since $q_{R}$ and $k_{R}$ have been previously determined [43], we omit these when considering the identifiability of our model [39]. The remaining parameters are all unidentifiable. Thus, we conclude that model 1 is structurally unidentifiable.

We carry out the same analysis for model 2 and find a similar trend. Disassociation rates for complexes *R*:*D*, *V*:*D*, *R*:*P* and *V*:*P* are all locally identifiable, as well as $k_{B}$, $q_{B}$ and $n_{B}$. The rest of the parameters of this model are unidentifiable, so that model 2 is unidentifiable too. Finally, and for model 3, making use of SIAN and the methods of Ref. [41] we find that all its parameters are locally identifiable. Therefore while carrying out parameter calibration it is important to keep in mind that with the data at hand, for models 1 and 2 we may limit what we may learn from the posterior distributions of their parameters. On the other hand, for model 3, since all parameters are locally identifiable we should be able to characterise the posterior distributions from this limited data set within some neighbourhood of the parameter space.

### 3.2. Sensitivity Analysis

Given the limited experimental data available to parameterise our models, it is vital to understand the significance of each parameter on the corresponding model output. To this end we make use of Sobol sensitivity analysis [44]. We choose the parameter ranges in Table 2 and make use of the Python package ‘*SALib*’. For each model we generated $10^{4}\left(2N+2\right)$ parameter sets using Satelli sampler, where *N* is the number of parameters for the model being considered [44,45,46]. For models 1 (Figure 1) and model 2 (Figure 2) we will examine how a change in parameter values affects the output, phosphorylated TBK1 ($B^{🟉}$). For model 3 (Figure 3) we instead examine the effect of changes in parameter values on the output, phosphorylated PKR:RNA complexes (*A*:*D*).

Table 3 lists the total order Sobol sensitivity indices for each of the three proposed mathematical models. For models 1 and 2 our analysis indicates that the most important parameter is $n_{B}$, the total number of TBK1 molecules. Changes in this parameter result in significant fluctuations in the chosen model output. The TBK1 phosphorylation rate, $k_{B}$, exhibits a large Sobol index of 0.482 and 0.462 for models 1 and 2, respectively. The binding rates $k_{V}$ and $k_{R}$ along with their associated unbinding rates carry an insignificant contribution to variation in model output (index ≤0.1) for both models 1 and 2. A similar trend is observed for the binding and unbinding rates involving the protein PACT in model 2. When considering the sensitivities of model 3 we notice that unbinding rates are insignificant, with a low index (<0.1). Parameters $n_{A}$ and $n_{D}$ are the most important ones in model 3. These have indices of 0.868 and 0.698, indicating any change in these parameters results in large model output fluctuations. The remaining parameters have a roughly equal level of importance. We note that in all three cases the total number of molecules for each model output has the largest sensitivity index. In models 1 and 2 this is followed by the TBK1 phosphorylation rate $k_{B}$, while for model 3 it is total number of viral RNA, $n_{D}$. Figure 4, Figure 5 and Figure 6 illustrate the time evolution of the sensitivity indices for each model. As can be observed the two most important parameters for each model remain so for the entire time course. Only in models 1 and 2 the importance of $q_{B}$ increased with time. Other parameters show little variation over time in their index value. Figure 6 shows that the Sobol sensitivity indices for model 3 remain constant with respect to time.

### 3.3. Parameter Calibration

We have introduced three different mathematical models which require parameter calibration. Data presented in Table 1 with approximate Bayesian computation sequential Monte Carlo (ABC-SMC) will be used to calibrate our models [37]. The rates associated with RIG-I, $k_{R}$ and $q_{V}$, are fixed using values obtained from the literature [39]. VAP rates, $k_{V}$ and $q_{V}$, are chosen to remain within the value of its dissociation constant [26,27]. As described by Toni et al. we execute K=18 iterations of the ABC-SMC method with n=2500 accepted parameter sets [37]. Figures 7, 9 and 10 present posterior histograms from the final iteration along with the model median and the 95% credible intervals. Table 4, Table 5 and Table 6 present posterior median and mean values for each parameter, with 95% credible interval also reported. For model 1 we fit $n_{B^{*}}$, for model 2 $n_{B^{*}}$ and for model 3 we fit $n_{AD}$.

Posterior histograms in Figure 7 for model 1 illustrate our inference can characterise the value $q_{B}$, with narrow posterior compared to its prior distribution. This is particularly important since sensitivity analysis indicated this was the third most important parameter, as shown in Table 3. We also obtain narrower posterior distributions for $k_{V}$, $q_{V}$ and $n_{V}$. Sensitivity analysis indicated that $k_{B}$ and $n_{B}$ were the two most important parameters to control within our model. However if we examine the correlations between these two variables we find a strong negative correlation between them (r = −0.83). Therefore, we will only be able to learn about their ratio. There also exists a positive correlation (r=0.65) between $n_{D}$ and $n_{V}$ as shown in Figure 8, so that with the data set and Bayesian inference, we can learn about the ratio of these two parameters. Upon re-examination of the issue of parameter identifiability when we allow the ratio of $n_{D}$ to $n_{V}$ to be constant, and making use of SIAN, we find the model becomes identifiable. We could therefore fix this ratio to obtain better estimates of our parameter values. This is out of the scope of this paper and has not been carried out.

Our Bayesian inference analysis for model 2 (see Figure 9) clearly indicates that very little is learnt for most parameters, including the rates associated with PACT binding to VAP, $k_{M}$, and its corresponding disassociation rate, $q_{M}$. Sensitivity analysis revealed the rates for TBK1 phosphorylation $k_{B}$ and de-phosphorylation $q_{B}$ along with $n_{B}$, the number of TBK1 molecules, are important to minimize variation in model output. Inference of the aforementioned parameters indicates narrow posterior histograms, which illustrates Bayesian inference is allowing us to learn about these three parameters. However, it is important to note the median number of molecules $n_{B}$ is low, which leads us to question its biological realism. Medians presented in Figure 7 and Figure 9 have a comparable trend. We note that model 2, additionally, has a rather sharp jump at the start of the time course. This is most likely due to the inclusion of PACT as a secondary activator. Importantly, the credible intervals of model 1 are narrower than those of model 2.

Figure 10 presents results from the Bayesian inference with model 3. The median course is similar to those in Figure 7 and Figure 9. Compared to model 2, the time course seems to better describe the data. In contrast, the median of model 1 is similar but has a smaller credible interval. When we examine the posterior distributions, we can see improve learning. Parameters for the number of viral RNA $n_{D}$ and PKR molecules $n_{A}$ have rather narrow posterior distributions compared to their prior ones. As mentioned in previous sections, these are rather important parameters as identified by Sobol sensitivity analysis. Thus, such improved learning is a rather desired feature of model 3, when compared to models 1 and 2. Many parameters have narrower posterior distributions when compared to their prior ones, which indicates overall learning for most parameter values. The rates $k_{M}$ and $q_{M}$ for the binding and unbinding of VP35 and PACT have the widest posterior distributions. Both rates could benefit from additional iteration steps in the of ABC-SMC method. Strong correlations exist particularly between $k_{M}$ and $n_{A}$ (r=−0.70) and $n_{P}$ (r=−0.71), which means we may only learn about the ratios of these values. Figure 11 illustrates a positive correlation between $q_{M}$ and $n_{V}$ (r=0.60), once again indicating learning about their ratio.

### 3.4. Model Selection

We have proposed three separate mathematical models, determined the sensitivity of their associated parameters, and used Bayesian inference to calibrate each model. We have also assessed their structural identifiability. Our initial findings would indicate that either model 1 or 3 would be suitable to appropriately describe the data, since they have the best posterior histograms and overall time course. We can therefore make use of ABC model selection and calculate the Akaike information criterion (AIC) to quantify which model better describes the data, and thus, which mechanism of type I IFN inhibition is preferred [47]. The second order AIC for small sample sizes is defined as
$$AIC_{c}=-2log\left(L\left(\theta \right)\right)+2K_{\theta }\left(\frac{n_{s}}{n_{s}-K_{\theta }-1}\right)\;,$$
with L(θ) the log-likelihood given parameters θ, defined in Section 2.3, $K_{\theta }$ defined as the number of estimated parameters for a given model, and $n_{s}$(=5) the number of samples used to generate the data presented in Table 1. The AIC value generated will be compared for each model, with a lower index being an indication of preferential model selection. We use a standard ABC rejection method, which unlike the ABC-SMC algorithm described previously, does not perform successive iterations. Here we define a threshold value, ε, fixed, and we accept parameters such that d<ε, where *d* is a distance measure. Here we use the Euclidean distance defined in Section 2.3. We run the ABC rejection method with ε=3 and count how many sampling instances are required to accept a total of $10^{5}$ parameter sets. Table 7 summarises our results.

Models 1 and 2 both have similar acceptance percentages: 5.35% of parameters sets accepted for model 1 and 5.12% for model 2. Model 3 has the largest percentage of accepted parameter sets at 9.03%, much higher than those by models 1 and 2. We note that model 3 has more parameters than models 1 and 2. Since we have a small number of data points and a large number of parameters, we make use of the second order AIC [47]. This method gives us AIC values of −19.25, −15.75 and −15.80 for models 1, 2 and 3, respectively. This shows that accounting for the number of parameters model 1 is the best, followed by model 3 then 2. Taking into account the ABC rejection results, we conclude that model 2 characterises the data poorly, and as such is less suitable than models 1 and 3. We now carry out a pair-wise comparison between models 1 and 3. We find a probability of 0.372 for choosing model 1, and of 0.628 for model 3. Thus, we conclude that Bayesian model selection indicates model 3 better explains the given limited data available.

## 4. Discussion

The ability of highly pathogenic viruses, such as Ebola virus, SARS-CoV-2 or CCHFV, to subvert innate immune responses results in severe infection and high fatality rates. The current pandemic has tragically illustrated the need to improve our understanding of the strategies viruses make use of to evade immune surveillance [48]. Of special relevance to viral infection are a family of cytokines: type I IFNs. Type I IFN responses (their expression induced by viral infection or the signalling cascades induced by their binding to specific IFN receptors) are frequently antagonised by viruses due to their importance in the initiation of innate immune responses [8]. Therefore, we believe it is timely and useful to develop quantitative approaches to characterise and quantify these evasion mechanisms, in a way that can be applied to different viruses.

In this manuscript we have focused on viral strategies to antagonise cytosolic type I IFN secretion pathways via PRRs [3], and have proposed three potential mathematical models with both upstream and downstream type I IFN inhibition mechanisms. These models have been formulated based on our current biological understanding of the interactions between the intra-cellular proteins involved [8,26]. In particular, we have investigated the role of a family of intra-cellular receptors, RLRs, which detect viral RNAs and promote IFN responses. VAPs may perturb a given RLR signalling pathway in its ‘upstream’ portion, at the level of dsRNA recognition, by competing with RLRs for dsRNA binding, or by removing PAMP signatures recognised by RLRs. Viral proteins such as NS1 (Influenza A), VP35 (Ebola), and N (SARS-COV), all bind viral RNAs, thus inhibiting PRRs from binding and signalling [8,26,27,49,50]. PAMPs such as NP (Lassa Fever) and NSP14 (SARS-COV) are removed, preventing RLR recognition. ‘Downstream’ effects include the inhibition, mediated by VAPs, of RLR-induced antiviral proteins [3,8]. For instance, VAPs may modify binding sites of proteins, inhibit formation of signalling complexes, or prevent translocation and phosphorylation [8]. Specific examples include the SARS-COV M protein which binds TRAF3 to form a long-lived complex, impeding its association with TBK1 and IKK-ε kinases to forge a functional signalling complex [51]. In the case of EBOV, its protein VP35 acts as a competing molecule for TBK1/IKK-ε with interferon regulatory factors IRF3 and IRF7, but also prevents their translocation to the cellular nucleus [52].

Mathematical models have been previously proposed to model interferon inhibition by viruses [14,15,16] or to describe inter-cellular interactions via IFN-α receptors [17]. These models are either virus specific or require detailed knowledge of many protein–protein interactions along the signalling pathway under consideration. Our aim in this manuscript is to characterise key biological hypotheses in a quantitative fashion, while avoiding, in principle, unnecessary complexity. Since clinical data sets from early viral infections are typically limited, it is important to have mathematical models in place that can be parameterised given this severe restriction, while our proposed models do not include all possible mechanisms of viral protein antagonism and inhibition of signalling pathways which result in type I IFN induction, the three models presented are a good first approximation. Moreover these models can be generalised to account for other mechanisms, proteins, or additional signalling pathways.

We proposed three separate models for the inhibition of type I interferon expression by VAPs. Each model considers a different biological mechanism or an alternate signalling pathway. Figure 1, Figure 2 and Figure 3 represent mechanisms which have been recently proposed in the literature [3,15,27,49]. For each model we have assessed its sensitivity and parameter identifiability, as well as carried out model selection and parameter calibration. In particular, we have made use of Sobol sensitivity analysis to identify, for each model and its output, which parameters would need to be closely controlled. We found that two parameters in each model need to be well characterised to avoid large variations in our model outputs. For models 1 and 2 these are the total number of TBK1 molecules, $n_{B}$, and its activation rate, $k_{B}$. In the case of model 3, the most important parameters are the total number of PKR molecules, $n_{A}$, and the total copy number of viral RNA molecules, $n_{D}$.

Unfortunately very little is known about the values for the parameters considered in our models. Thus, our aim was to carry out Bayesian inference to narrow down these values. To this end, it was also important to carry out a structural identifiability analysis. This analysis led to the following results: model 3 (considering the PKR pathway) is locally identifiable, but models 1 and 2 are not. We note that these results are in light of the limited data set we had at hand. Yet, this indicated that even with limited data, model 3 might be better, when compared to the other two models, at allowing us to infer parameter values. This was further supported by model selection and parameter inference [37,41]: the PKR signalling pathway has a higher percentage of acceptance as illustrated in Table 7 and narrower posterior distributions for most parameter values (see Figure 10). Overall model 2 was deemed the worst model of the three: many parameters were non-identifiable, it led to the worst percentage of parameter set acceptance from model selection, and wide posterior distributions for its parameters. Model 1 cannot be rejected since it had the lowest AIC coefficient. However as previously mentioned this model leads to poor learning for most parameter values and is structurally unidentifiable.

While the deterministic models we have presented can be generalised to be applicable to other viruses, it should be remembered that there exist additional signalling pathways and molecular mechanisms which have not been included in the models. Ebola virus infection, which we have examined as a case study, has a number of specific features not characterised by the mechanisms included in our models. Plasmacytoid dendritic cells (pDCs) have been shown to be refractory to EBOV infection, where as common dendritic cells are susceptible [53]. This could be due to the fact that pDCs express basal (or constitutive) levels of IRF7 prior to infection [54], and therefore, can be considered in an antiviral state. Thus, when considering the development of a mathematical model it is important to understand not only the virus but also the cellular tropism of the virus and the host (i.e., invertebrate or vertebrate); that is, which cells are the target cells of the virus [55]. Here we have focused on type I IFNs as essential antiviral cytokines, yet immune responses require a complex and coordinated interaction of a large collection of cytokines and cells, which are out of the scope of this paper. We argue that a more comprehensive data set will be required to consider the development of mathematical models of such complexity [56,57].

A final point to consider is the difference between in vivo and in vitro infection. Our models have been parameterised with an in vivo data set. It has been recently highlighted that there exists a stark contrast in responses when comparing in vivo and in vitro infection; in particular, and for the in vitro case, type I IFN production is abrogated after three days post-infection, whereas in the case of in vivo infection type I IFN is present throughout the entire infective period [11,12]. Hence it is critical to keep this in mind when carrying out parameter calibration. Along this line of thought, it is also important to note that differences in in vivo experimental models can lead to rather different innate immune responses. For instance, bats are a proposed reservoir for Ebola virus but are known to be asymptomatic for disease. Experiments have indicated that bats have detectable viral RNA levels, but no detectable viremia [58]. Yet, in the case of humans and non-human primates, the clinical presentation tends to be symptomatic and with measurable viremia [59,60]. Bat dendritic cells have shown an enhanced capacity to initiate IFN-dependent responses upon filovirus infection in comparison with, for example, human cells [61]. Other studies have reported a difference in immune responses depending on the specific tissue analysed: Ebola viral RNA levels persist in male gonads even after a negative PCR test from blood samples [62].

We believe the mathematical approaches presented in this manuscript have allowed us to explore different mechanisms of viral antagonism of type I IFN production. These models could be further expanded to incorporate other intra-cellular or viral mechanisms, or additional signalling pathways. Additional data sets, e.g., from quantitative proteomics, could be used to improve parameter inference, not only for Ebola virus but for other viruses which are of global concern and a public health threat (see Figure 12). A general (or abstract) model of viral type I IFN inhibition should consider the following proteins and molecules: *V*, the viral antagonistic protein for the virus under consideration; *D*, the viral RNA; $R_{1}$, a PRR; *P*, a dsRNA binding protein; $R_{2}$, an RNA-activated kinase; and *E*, a downstream enzyme kinase. The model can be described by the following reactions, which assume mass action kinetics:A pattern recognition receptor, $R_{1}$ binds to *D* to form $R_{1}:D$, with forward rate $k_{R_{1}}$ and backward rate $q_{R_{1}}$. This is the first step in the $R_{1}$ signalling pathway.$R_{1}$ binds to *P* to form $R_{1}$:*P*, with forward rate $k_{P}$ and backward rate $q_{P}$.*V* binds to *D* to form *V*:*D*, with forward rate $k_{V}$ and backward rate $q_{V}$. This is an upstream inhibition mechanisms: *V* sequesters *D* from its potential binding to $R_{1}$.*V* binds to *P* to form *V*:*P*, with forward rate $k_{M}$ and backward rate $q_{M}$. This is a second upstream inhibition mechanisms: *V* sequesters *P* from its potential binding to $R_{1}$.Downstream enzyme kinase, *E*, gets phosphorylated in the presence of complexes $R_{1}$:*P* and $R_{1}$:*D*. This phosphorylation event is inhibited by the presence of *V*. This encodes downstream inhibition by *V*. Phosphorylation and de-phosphorylation rates of *E* and $E^{🟉}$ are $k_{E}$ and $q_{E}$, respectively.A second intra-cellular signalling pathway can be considered, in addition to the one initiated by $R_{1}$. To this end, we consider an RNA-activated kinase, $R_{2}$, which can bind to *D* to form the signalling complex $R_{2}$:*D*. A final mechanism of inhibition by *V* is included in this pathway. *V* enhances the disassociation of $R_{2}$:*D*.

One last note to conclude. Our models have been restricted to be of a deterministic nature and not stochastic. We plan to generalise the models introduced here to birth and death Markov processes [63] to include the effects that might arise due to a small number of proteins, as can be the case during the first stages of intra-cellular infection. Effects which have been completely ignored in our models, since we have assumed the initial number of molecules ($n_{B},n_{D},n_{R},n_{V}$ and $n_{A}$) is large to allow us for a deterministic approach. This might not be the case always, and thus, an stochastic perspective will be desirable and required.

## Figures and Tables

**Figure 1 viruses-13-02441-f001:**
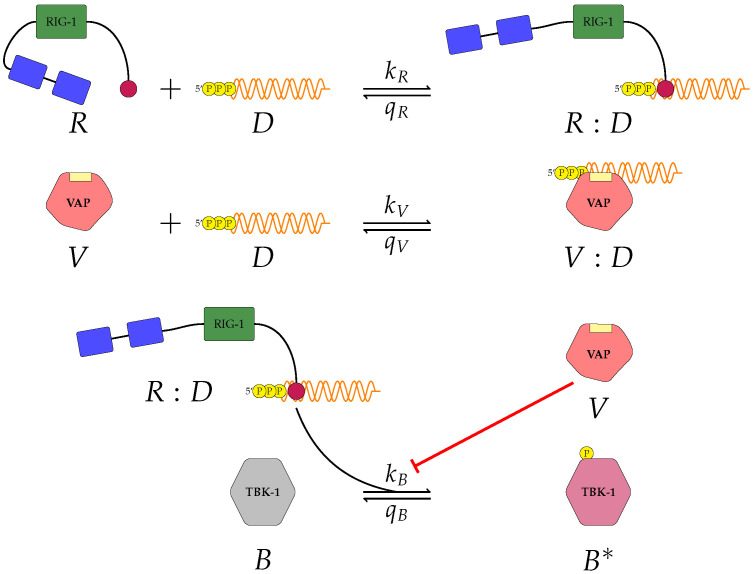
A first model of type I IFN inhibition by VAP (model 1). Model 1 considers the following molecules: RIG-I denoted *R*, viral RNA denoted *D*, VAP denoted *V*, and TBK1 denoted *B*. In this model there are six reactions and three molecular complexes. Free VAP inhibits phosphorylation of TBK1, as indicated by the flat head red arrow.

**Figure 2 viruses-13-02441-f002:**
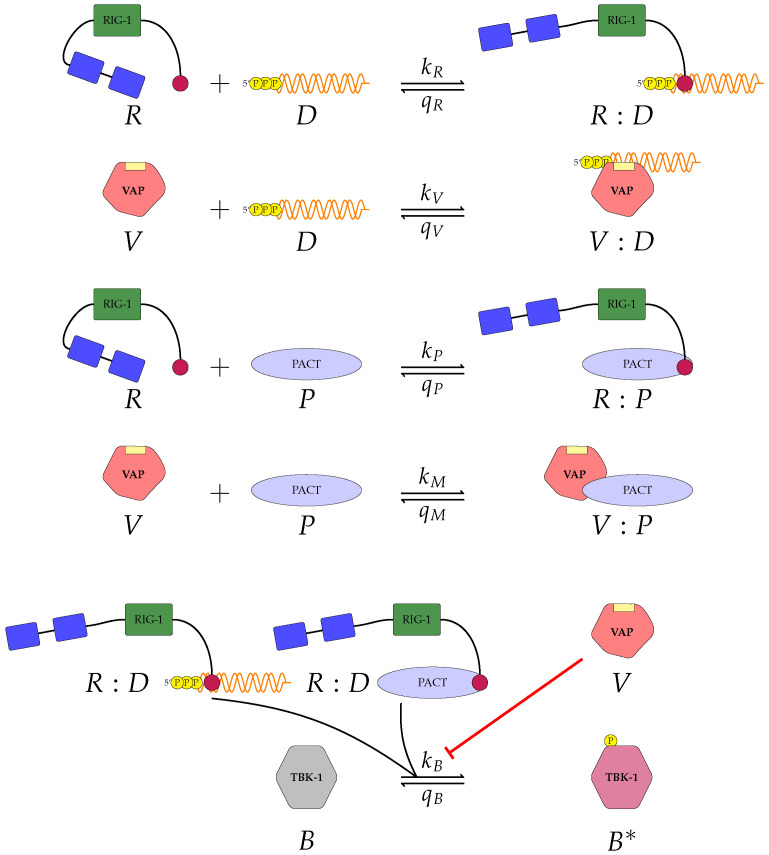
A second model of type I IFN inhibition by viral protein (model 2). Model 2 includes the role of protein activator of the interferon-induced protein kinase (PACT) molecules, since PACT has been identified as a secondary activator of RIG-I during viral infections [23,28]. In this model there are ten reactions and five molecular complexes. Free VAP inhibits phosphorylation of TBK1, as indicated by the flat head red arrow.

**Figure 3 viruses-13-02441-f003:**
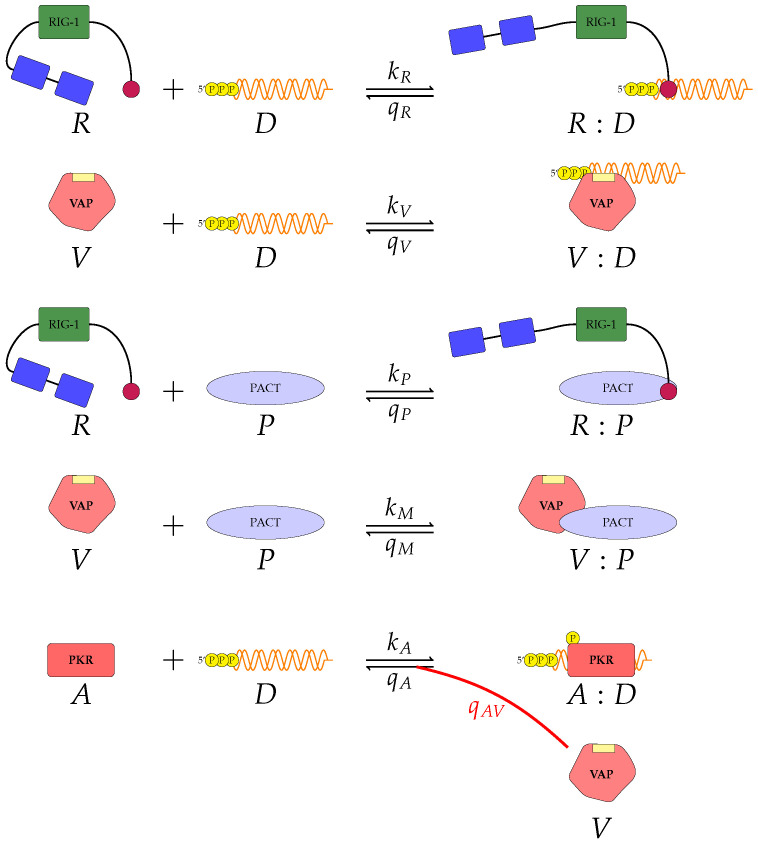
A third model of type I IFN inhibition by viral protein (model 3). It includes the contribution of the PKR pathway. In this model there are ten reactions and five molecular complexes. In the presence of free VAP, phosphorylated PKR is actively de-phosphorylated, as indicated by the red reverse reaction arrow with rate $q_{AV}$.

**Figure 4 viruses-13-02441-f004:**
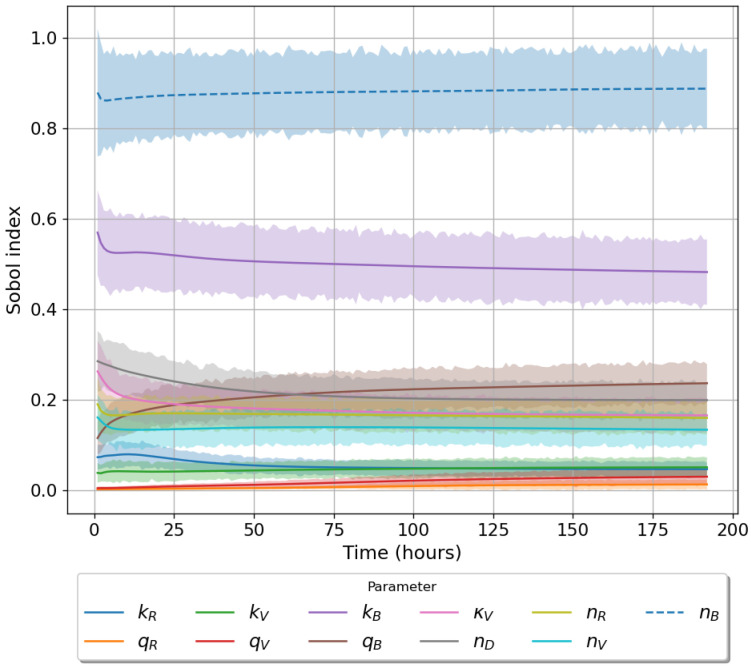
Time evolution of total order Sobol sensitivity indices (model 1). Model output for this model is activated TBK1, $n_{B^{*}}$. Shaded region accounts for a 95% confidence interval.

**Figure 5 viruses-13-02441-f005:**
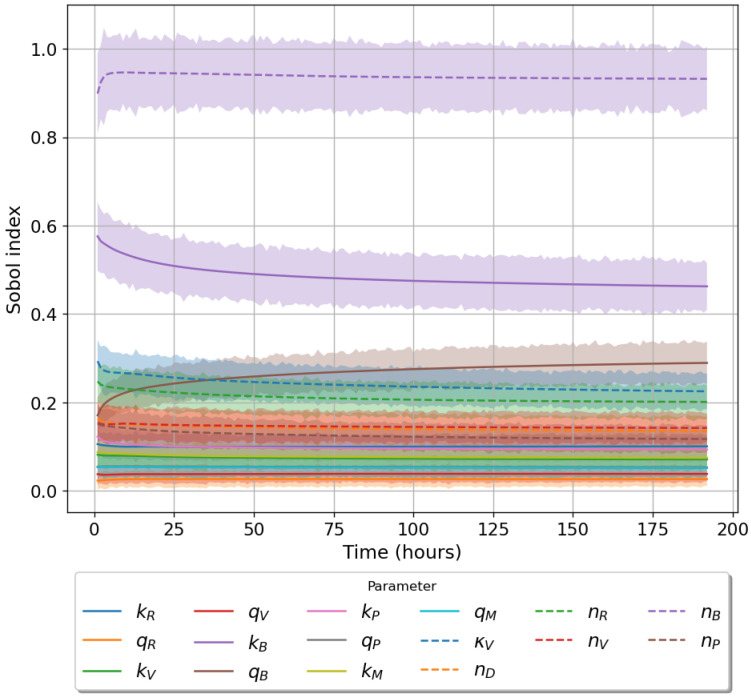
Time evolution of total order Sobol sensitivity indices (model 2). Model output for this model is activated TBK1, $n_{B^{*}}$. Shaded region accounts for a 95% confidence interval.

**Figure 6 viruses-13-02441-f006:**
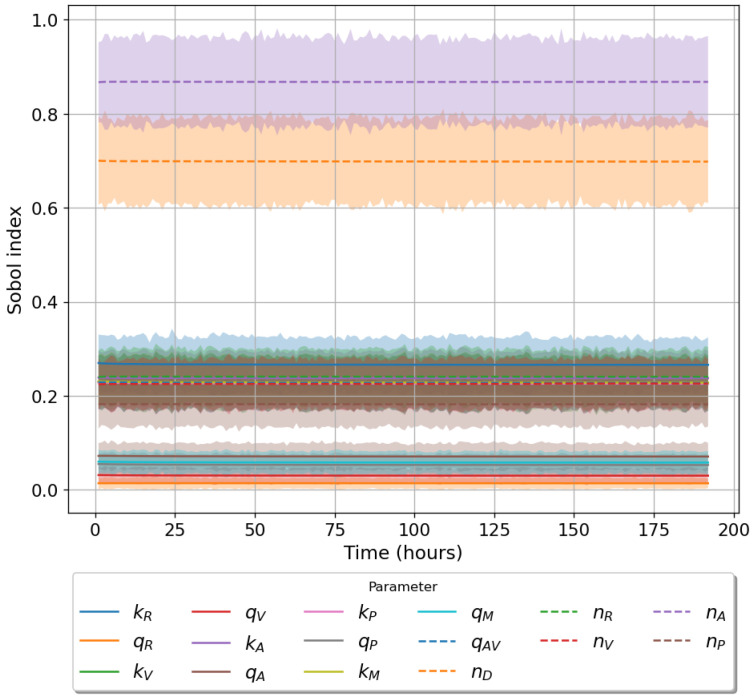
Time evolution of total order Sobol sensitivity indices (model 3). Model output for this model is phosphorylated PKR, $n_{AD}$. Shaded region accounts for a 95% confidence interval.

**Figure 7 viruses-13-02441-f007:**
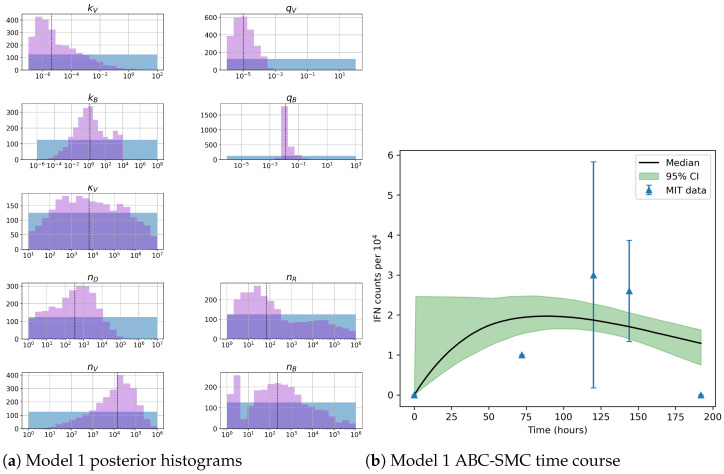
(**a**) Posterior histograms obtained from the ABC-SMC algorithm. Blue histograms indicate prior distributions and purple histograms illustrate posterior ones. (**b**) Model fit from accepted parameter sets obtained during the final iteration of the ABC-SMC algorithm. Blue triangles represent data presented in Table 1 plotted with its standard deviation. The black line illustrates the point-wise median value from the accepted parameter sets (shaded in green) with a 95% credible interval. These results represent 18 iterations with 2500 accepted parameter sets for model 1.

**Figure 8 viruses-13-02441-f008:**
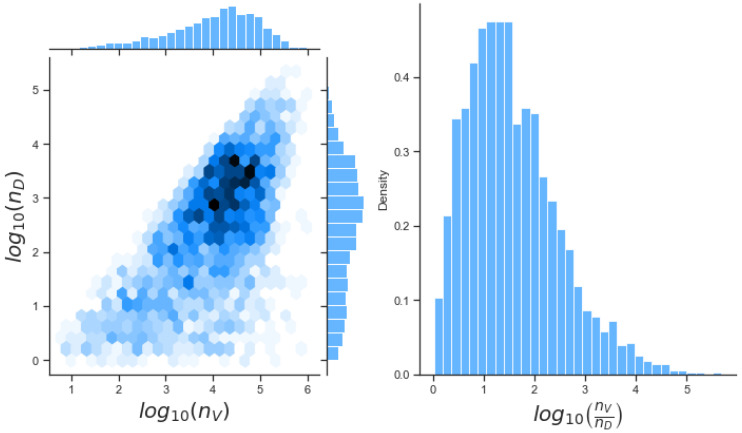
Left: bivariate posterior histogram of $\log_{10}\left(n_{D}\right)$ and $\log_{10}\left(n_{V}\right)$ showing a positive correlation (r=0.65). Right: posterior distribution for the ratio $\log(n_{V}/n_{D})$.

**Figure 9 viruses-13-02441-f009:**
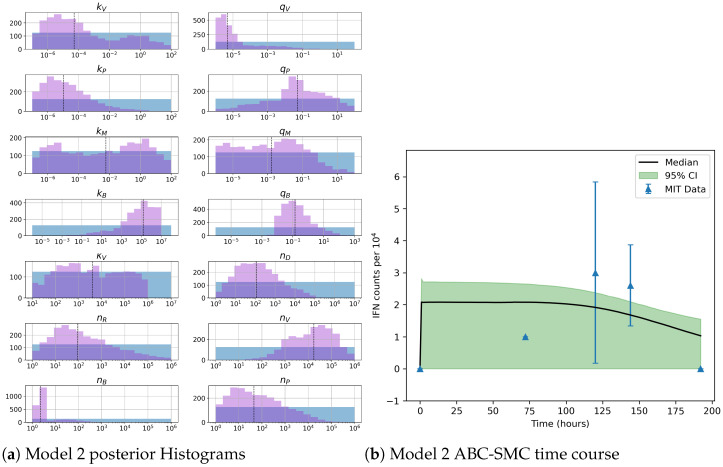
(**a**) Posterior histograms obtained from the ABC-SMC algorithm. Blue histograms indicate prior distributions and purple histograms illustrate posterior ones. (**b**) Model fit from accepted parameter sets obtained during the final iteration of the ABC-SMC algorithm. Blue triangles represent data presented in Table 1 plotted with its standard deviation. The black line illustrates the point-wise median value from the accepted parameter sets (shaded in green) with a 95% credible interval. These results represent 18 iterations with 2500 accepted parameter sets for model 2.

**Figure 10 viruses-13-02441-f010:**
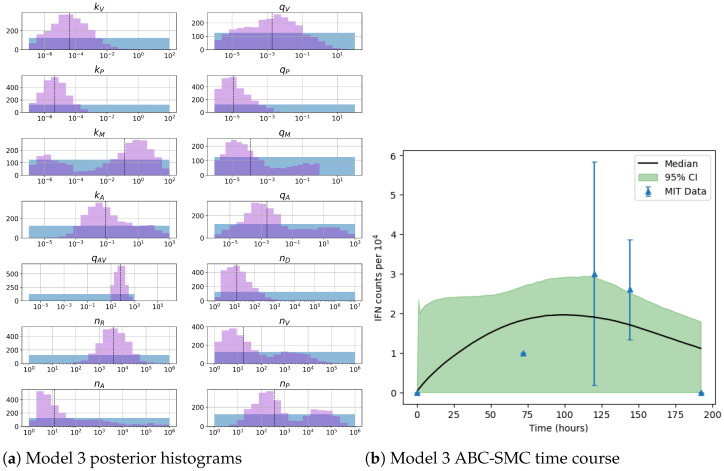
(**a**) Posterior histograms obtained from the ABC-SMC algorithm. Blue histograms indicate prior distributions and purple histograms illustrate posterior ones. (**b**) Model fit from accepted parameter sets obtained during the final iteration of the ABC-SMC algorithm. Blue triangles represent data presented in Table 1 plotted with its standard deviation. The black line illustrates the point-wise median value from accepted parameter sets (shaded in green) with a 95% credible interval. These results represent 18 iterations with 2500 accepted parameter sets for model 3.

**Figure 11 viruses-13-02441-f011:**
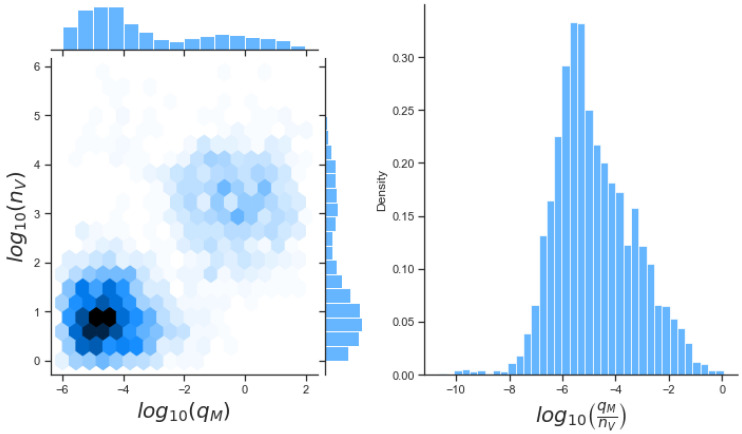
Left: bivariate posterior histogram of $\log_{10}\left(q_{M}\right)$ and $\log_{10}\left(n_{V}\right)$ showing a positive correlation (r=0.60). Right: Posterior distribution of the ratio $\log(q_{M}/n_{V})$.

**Figure 12 viruses-13-02441-f012:**
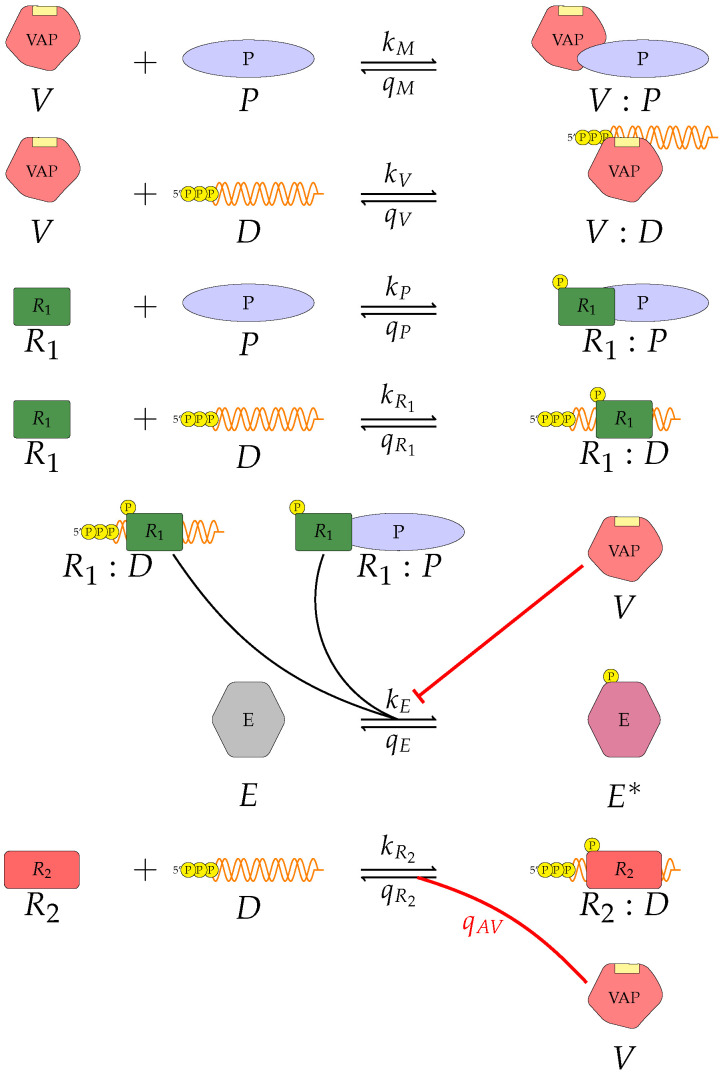
A general model of viral type I IFN inhibition. The model includes: *V*, the viral antagonistic protein for the virus under consideration; *D*, the viral RNA; $R_{1}$, a PRR; *P*, a dsRNA binding protein; $R_{2}$, an RNA-activated kinase; and *E*, a downstream enzyme kinase. The model includes two potential signalling pathways that result in stimulation of type I IFN secretion.

**Table 1 viruses-13-02441-t001:** Mean transcript counts (with standard deviation) for IFN-β from a longitudinal study of EBOV infection in rhesus macaques [12].

Day	Mean Counts per $10^{4}$ (SD)
0	0 (0)
3	1 (0)
5	3 (2.83)
6	2.6 (1.26)
8	0 (0)

**Table 2 viruses-13-02441-t002:** Parameter ranges used for Bayesian inference. Ranges were taken based on the published and data, with two to three orders of magnitude taken on either side of these values [11,15,26,27,39]. We consider uniform distributions of exponent base 10 for all prior choices. We then restrict parameters $k_{V}$ and $q_{V}$ by making sure their ratio, $\frac{k_{V}}{q_{V}}$, lies within the dissociation constant range 3.40 μM–1.1 nM reported in Refs. [26,27].

Parameter	Search Range	Units	References	Model
$k_{R}$	0.04836	$h^{-1}$ per molecule pair	[39]	1, 2, 3
$q_{R}$	7.632	$h^{-1}$ per molecule	[39]	1, 2, 3
$k_{V}$	U[−7,2]	$h^{-1}$ per molecule pair	[26,27]	1, 2, 3
$q_{V}$	U[−6,2]	$h^{-1}$ per molecule	[26,27]	1, 2, 3
$k_{P}$	U[−7,−2]	$h^{-1}$ per molecule pair		2, 3
$q_{P}$	U[−6,0]	$h^{-1}$ per molecule		2, 3
$k_{M}$	U[−7,−2]	$h^{-1}$ per molecule pair		2, 3
$q_{M}$	U[−6,0]	$h^{-1}$ per molecule		2, 3
$k_{B}$	U[−6,8]	$h^{-1}$ per molecule	[15,40]	1, 2
$q_{B}$	U[−6,3]	$h^{-1}$ per molecule	[15,40]	1, 2
$k_{A}$	U[−7,2]	$h^{-1}$ per molecule pair		3
$q_{A}$	U[−6,2]	$h^{-1}$ per molecule		3
$q_{AV}$	U[−6,2]	$h^{-1}$ per molecule		3
$\kappa_{V}$	U[1,7]			1, 2
$n_{D}$	U[0,7]			1, 2, 3
$n_{R}$	U[0,6]			1, 2, 3
$n_{V}$	U[0,6]			1, 2, 3
$n_{B}$	U[0,6]			1, 2
$n_{A}$	U[0,6]			3

**Table 3 viruses-13-02441-t003:** Total order Sobol sensitivity indices for each proposed model. Parameters are listed from most important to least, according to sensitivity index [44]. $10^{4}$ samples were generated with sensitivity to $n_{B^{*}}$ for model 1, $n_{B^{*}}$ for models 2 and $n_{AD}$ for model 3.

	Model 1		Model 2		Model 3	
	Parameter	Index	Parameter	Index	Parameter	Index
Most important	$n_{B}$	0.887	$n_{B}$	0.932	$n_{A}$	0.868
	$k_{B}$	0.482	$k_{B}$	0.462	$n_{D}$	0.698
	$q_{B}$	0.236	$q_{B}$	0.289	$k_{R}$	0.266
	$n_{D}$	0.199	$\kappa_{V}$	0.225	$n_{R}$	0.241
	$\kappa_{V}$	0.165	$n_{R}$	0.201	$k_{A}$	0.238
	$n_{R}$	0.159	$n_{V}$	0.142	$k_{M}$	0.228
	$n_{V}$	0.133	$n_{D}$	0.135	$q_{AV}$	0.227
	$k_{V}$	0.050	$n_{P}$	0.117	$n_{V}$	0.226
	$k_{R}$	0.046	$k_{R}$	0.100	$k_{V}$	0.226
	$q_{V}$	0.029	$k_{P}$	0.093	$k_{P}$	0.226
	$q_{R}$	0.012	$k_{M}$	0.074	$n_{P}$	0.182
			$k_{V}$	0.071	$q_{A}$	0.07
			$q_{P}$	0.053	$q_{M}$	0.058
			$q_{M}$	0.051	$q_{P}$	0.053
			$q_{V}$	0.038	$q_{V}$	0.030
Least important			$q_{R}$	0.026	$q_{R}$	0.014

**Table 4 viruses-13-02441-t004:** Summary statistics for each accepted parameter value sets from model 1. Mean, median and a 95% credible interval are summarised in the table.

Parameter	Median	Mean	Credible Interval
$k_{V}$	$4.37\times 10^{-6}$	$2.04\times 10^{-2}$	$\left[1.42\times 10^{-7},5.58\times 10^{-2}\right]$
$q_{V}$	$1.03\times 10^{-5}$	$3.42\times 10^{-5}$	$\left[1.35\times 10^{-6},2.22\times 10^{-4}\right]$
$k_{B}$	1.50	$4.69\times 10^{2}$	$\left[5.78\times 10^{-4},5.62\times 10^{-3}\right]$
$q_{B}$	$1.29\times 10^{-2}$	$2.74\times 10^{-2}$	$\left[6.24\times 10^{-3},1.49\times 10^{-1}\right]$
$\kappa_{V}$	$6.70\times 10^{3}$	$374\times 10^{5}$	$\left[2.00\times 10^{1},4.08\times 10^{6}\right]$
$n_{D}$	$3.37\times 10^{2}$	$3.34\times 10^{3}$	$\left[1,2.92\times 10^{4}\right]$
$n_{R}$	$6.80\times 10^{1}$	$2.82\times 10^{4}$	$\left[1,3.67\times 10^{5}\right]$
$n_{V}$	$1.39\times 10^{4}$	$4.38\times 10^{4}$	$\left[5.2\times 10^{1},2.69\times 10^{5}\right]$
$n_{B}$	$2.26\times 10^{2}$	$1.91\times 10^{4}$	$\left[1,2.54\times 10^{5}\right]$

**Table 5 viruses-13-02441-t005:** Summary statistics for each accepted parameter value sets from model 2. Mean, median and a 95% credible interval are summarised in the table.

Parameter	Median	Mean	Credible Interval
$k_{V}$	$5.19\times 10^{-5}$	1.38	$\left[1.96\times 10^{-7},1.52\times 10^{1}\right]$
$q_{V}$	$5.04\times 10^{-6}$	$6.36\times 10^{-2}$	$\left[1.07\times 10^{-6},4.81\times 10^{-2}\right]$
$k_{P}$	$1.05\times 10^{-5}$	$3.45\times 10^{-2}$	$\left[1.60\times 10^{-7},1.00\times 10^{-1}\right]$
$q_{P}$	$5.06\times 10^{-2}$	3.03	$\left[1.02\times 10^{-5},3.44\times 10^{1}\right]$
$k_{M}$	$6.08\times 10^{-3}$	3.27	$\left[2.12\times 10^{-7},3.89\times 10^{1}\right]$
$q_{M}$	$1.69\times 10^{-3}$	$6.83\times 10^{-1}$	$\left[1.45\times 10^{-6},4.949\right]$
$k_{B}$	$1.55\times 10^{5}$	$4.75\times 10^{6}$	$\left[1.134,4.95\times 10^{7}\right]$
$q_{B}$	$1.41\times 10^{-1}$	3.29	$\left[8.75\times 10^{-3},27.86\right]$
$\kappa_{V}$	$3.96\times 10^{3}$	$2.67\times 10^{5}$	$\left[1.7\times 10^{1},2.68\times 10^{6}\right]$
$n_{D}$	$1.14\times 10^{2}$	$2.64\times 10^{3}$	$\left[2,2.59\times 10^{4}\right]$
$n_{R}$	$9.2\times 10^{1}$	$1.36\times 10^{4}$	$\left[1,1.36\times 10^{5}\right]$
$n_{V}$	$1.76\times 10^{4}$	$6.74\times 10^{4}$	$\left[1.98\times 10^{2},4.57\times 10^{5}\right]$
$n_{B}$	2	$2.61\times 10^{3}$	$\left[1,2.97\times 10^{2}\right]$
$n_{P}$	$4.5\times 10^{1}$	$1.98\times 10^{3}$	$\left[1,1.72\times 10^{4}\right]$

**Table 6 viruses-13-02441-t006:** Summary statistics for each accepted parameter value sets from model 3. Mean, median and a 95% credible interval are summarised in the table.

Parameter	Median	Mean	Credible Interval
$k_{V}$	$3.87\times 10^{-5}$	$3.00\times 10^{-2}$	$\left[2.84\times 10^{-7},1.40\times 10^{-2}\right]$
$q_{V}$	$1.92\times 10^{-3}$	$2.36\times 10^{-1}$	$\left[2.43\times 10^{-6},1.66\right]$
$k_{P}$	$4.55\times 10^{-6}$	$7.74\times 10^{-2}$	$\left[1.79\times 10^{-7},3.15\times 10^{-4}\right]$
$q_{P}$	$1.26\times 10^{-5}$	$3.20\times 10^{-3}$	$\left[1.22\times 10^{-6},1.38\times 10^{-3}\right]$
$k_{M}$	$1.30\times 10^{-1}$	3.30	$\left[2.61\times 10^{-7},2.90\times 10^{1}\right]$
$q_{M}$	$1.14\times 10^{-4}$	1.56	$\left[1.72\times 10^{-6},1.81\times 10^{1}\right]$
$k_{A}$	$8.41\times 10^{-2}$	$2.30\times 10^{1}$	$\left[6.54\times 10^{-4},2.80\times 10^{2}\right]$
$q_{A}$	$2.32\times 10^{-3}$	$1.71\times 10^{1}$	$\left[4.90\times 10^{-6},2.07\times 10^{2}\right]$
$q_{AV}$	$6.31\times 10^{1}$	$9.32\times 10^{1}$	$\left[1.25\times 10^{1},3.82\times 10^{2}\right]$
$n_{D}$	$1.30\times 10^{1}$	$2.37\times 10^{3}$	$\left[2,5.98\times 10^{2}\right]$
$n_{R}$	$4.16\times 10^{3}$	$9.90\times 10^{3}$	$\left[3.24\times 10^{2},5.16\times 10^{4}\right]$
$n_{V}$	$1.70\times 10^{1}$	$3.46\times 10^{3}$	$\left[1,2.00\times 10^{4}\right]$
$n_{A}$	$1.20\times 10^{1}$	$1.61\times 10^{4}$	$\left[2,2.00\times 10^{5}\right]$
$n_{P}$	$3.74\times 10^{2}$	$2.19\times 10^{4}$	$\left[1.40\times 10^{1},1.76\times 10^{5}\right]$

**Table 7 viruses-13-02441-t007:** Table with number of sample parameter sets required for $10^{5}$ sets to be accepted. An Euclidean distance measure was used with ε=3 as threshold value for acceptance. Percentage of accepted parameter sets also shown.

	Number of Samples	% Accepted	Relative Probability
Model 1	1,868,652	5.35	0.274
Model 2	1,952,835	5.12	0.263
Model 3	1,107,525	9.03	0.463

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
