# Peer review of "Quantification of Type I Interferon Inhibition by Viral Proteins: Ebola Virus as a Case Study"

_viruses, 2021, doi:10.3390/v13122441_

Round 1

Reviewer 1 Report

The authors develop three molecular models for the interaction between viral proteins and their role in the inhibition of type 1 INF secretion and them to infer which model best describes data from ebola infection in rhesus macaques. The paper is well written and the methods are clearly presented. I have a few suggestions/concerns:

  1. The models in figures 1, 2 and 3 would benefit from feedbacks between equations. For example, can you have an inhibition type arrows coming into the phosphorylation of B from V?
  2. While data in figure 1 has only 5 data points, I believe the authors use 17 data points during data fitting. Can you explain how these additional points were generated?
  3. When writing the results, please specify which parameters are you referencing, rather than just calling them binding/unbinding rates.
  4. Is it possible to make model one identifiable by finding correlated parameters and fixing their product/ ratio as with nV and nD?
  5. What can you say about practical identifiability of model 3?
  6. Most importantly, what does excluding model two mean for ebola infection? Can you make biological inferences? Suggest interventions?

Author Response

Please, see attached file for our reply to reviewer one.

Reviewer 2 Report

Thank you for the opportunity to review this manuscript. I like the overall approach, and in particular the use of Approximate Bayesian Computation in order to identify plausible parameter spaces/sets. My comments are primarily focused on the selection of example models, and some of the assumptions related to those decisions. In order for me to better understand what each model is doing, I have tried to summarize them here (and for the authors to correct me if I misrepresented something):

Model 1: Competitive binding of VAP to vRNA to reduce amount of vRNA to activate RIG. Also some reduction of phosphorylation of TBK1 by free VAP (total VAP minus amount bound to vRNA)

Model 2: Same Model 1 Competitive binding VAP to vRNA to reduce amount of vRNA to activate RIG, also Competitive binding of VAP to PACT to reduce amount of PACT to activate RIG, same reduction of phosphorylation of TBK1 by free VAP (total VAP minus amount bound to vRNA and PACT)

Model 3: Same as Model 2 but instead of TBK1 use PKR, but instead of phosphorylation inhibition competitive binding by VAP, PKR binds to vRNA, which is competitively bound by VAP. So PKR is “inhibited” competitively by VAP, just as RIG is, so both RIG and PKR compete with each other (for vRNA) in addition with VAP, whereas PACT just competes with VAP.

My questions/comments are below:

  1. As presented, it appears that the authors are making the assumption that the various models are mutually exclusive by their use of the term “model selection”, even though they are all present in Ebola. Are the authors trying to say that despite being all present in the virus that they are trying to find the “single” one that explains the data best, even though in the real world there is going to be potential differential effects of all three? I apologize if I am missing something basic here, but the central premise is unclear to me.
  2. Potentially related to #1 is the appearance that Model 2 just adds an additional mechanism to Model 1, and therefore is more “real” than Model 1 because it incorporates two mechanisms known to be present in Ebola. Therefore, in comparing Model 1 to Model 2, where Model 1 appears to better fit the data, does that mean that the PACT pathway is not important?
  3. Similarly, is TBK1 assumed to be nonfunctional if one assumes Model 3, which has a completely different end phosphorylation target (PKR)? Does Model 3 therefore assume that TBK1 has no role?
  4. In the interest of developing a method that can be applied to different viruses (which the authors state is their goal), it seems like there are certain motifs (here all related to competitive binding) that are present in their 3 models, that, if described abstractly and combined to give the 3 models, might help someone (like myself) who can get confused with the specific molecular species names.
  5. The issue of model parsimony/complexity was brought up and there are interesting findings in this paper related to this. In one case, the simpler Model 1 has advantages over Model 2 but the most complex Model 3 appears to be the “best” one. This raises the question of some “middle zone” of complexity that is undesirable. Related to my prior comments perhaps there is some more general categorization of how model complexity is added that can suggest why this occurs? 
  6. I would ask why the authors target the mean value of the data? The SD bars in the middle two time points show a great deal of variance, it would seem like there is a great deal of potential information in how that variance was generated? This is particularly true as this is in vitro data, which is much more highly controlled and therefore the variance reflects “true” cellular heterogeneity (as opposed to confounding factors as may be present in vivo). I would be interested to hear the authors opinions on this observation.

Author Response

Please, see attached file for our reply to reviewer two.

Round 2

Reviewer 2 Report

I am satisfied with the responses to my prior comments.